# Ageing and heterogeneity regarding autism spectrum conditions: a protocol paper of an accelerated longitudinal study

Hilde M Geurts ,[1,2] Joost A Agelink van Rentergem ,[1] Tulsi Radhoe ,[1] Carolien Torenvliet ,[1] Wikke J Van der Putten ,[1,2] Annabeth P Groenman [1]

► Prepublication and additional material history for this paper is available online. To view these files, please visit the journal online (http://dx.doi.org/10.1136/bmjopen-2020-040943).

## ABSTRACT

**Introduction** Autism spectrum conditions (ASC) develop early in life and are thought to last a lifetime. However, ASC research has two major knowledge gaps that hinder progression in understanding the concept of ASC and in providing proper support for autistic adults: (1) the majority of knowledge about ASC mainly stems from childhood studies so little is known about older autistic adults and (2) while it is broadly recognised that ASC is a heterogeneous condition, we do not yet understand the differences in trajectories leading to their future outcome. We aim to fill both knowledge gaps.

**Methods and analysis** A multistage overlapping cohort design assessing (cognitive) ageing in ASC is designed to obtain an accelerated longitudinal data set. Data, including a multitude of questionnaires, diagnostics and cognitive tests, are collected over four waves within a 10-year time frame. This will provide information regarding actual changes in quality of life, co-occurring health conditions and cognition as well as the possibility to test external validity and temporal stability in newly formed behavioural subtypes. Participants consist of three groups of adults aged 20–90 years: (1) with a clinical diagnosis of ASC, (2) with a clinical diagnosis of attention deficit hyperactivity disorder (ADHD) but no ASC, (3) no ASC/ADHD (ie, comparison group). The sample size differs between waves and instruments. Detailed analysis plans will be preregistered in AsPredicted or at the Open Science Framework.

**Ethics and dissemination** Ethical approval for this study was obtained from the ethical review board of the Department of Psychology of the University of Amsterdam (wave 1 2011-PN-1952 and 2013-PN-2668, wave 2 2015-BC-4270, waves 3 and 4 2018-BC-9285). In line with the funding policies of the grant organisation funding this study, future papers will be published open access.

¹Psychology, University of Amsterdam, Amsterdam, The Netherlands
²Leo Kannerhuis (Youz), Amsterdam, The Netherlands

**Correspondence to**
Dr Hilde M Geurts;
H.M.Geurts@uva.nl

## Strengths and limitations of this study

► This study has an accelerated longitudinal design, so in a relative short time frame, for the first time, actual age-related (cognitive) changes can be assessed in older autistic adults.
► We will use in-depth phenotyping due to inclusion of measures of known vulnerability and protective factors regarding ageing and a wide range of diagnostic instruments, questionnaires and cognitive tests.
► The replicability, external validity and temporal stability of behavioural ASC subgroups can be tested, which are aspects that are often not implemented in subgroup study designs.
► The first two measurement waves were not set up as being part of an accelerated longitudinal design which led to high attrition rates.
► As such, not all measures are the same across all four measurement waves.

## INTRODUCTION

Neurodevelopmental conditions, like an autism spectrum conditions (ASC), are defined as conditions with early atypical brain development, which have cascading effects on cognitive processes important during the entire lifespan.[1] At the core of the Diagnostic and Statistical Manual of Mental Health Disorders Fifth Edition (DSM-5), ASC diagnoses are difficulties in social interaction and communication, sensory sensitivities and stereotyped or repetitive behaviours and interests.[1] A conservative estimate is that one in hundred persons meets the criteria,[2] but recent estimates seem to be higher and vary widely,[3] Whereas ASC was originally perceived as a childhood condition, which was often comorbid with a severe intellectual disability and had an overall poor outcome, we now know that ASCs are not restricted to childhood or to those with low intellectual functioning, and that there is large heterogeneity across persons and across time in type and severity of the disabilities experienced.[4 5] This led to the idea that while ASC is a behaviorally defined condition, it likely consists of different conditions with diverse outcomes that are potentially falsely lumped together in one diagnostic category.[6 7] Understanding the concept, course and consequences of ASCs is critical in order to develop appropriate

support for (old) autistic[i] adults and for adults with related neurodevelopmental conditions. Unfortunately, ASC research hardly exceeded young adulthood, so what happens in old age remained exempt of empirical scrutiny[8 9] (for an overview of reasons for the lack of ageing research[10 11]). Thus, two questions need to be answered:

1. How do people with ASCs progress in late adulthood?
2. How can we understand the differences among people with ASC in future outcome?

While the process of cognitive decline might be similar for autistic adults and nonautistic adults (ie, parallel ageing or delayed ageing where cognitive maturation and cognitive decline are shifted in time, but follow a similar pattern), a dominant assumption is that autistic adults are at a larger risk for accelerated cognitive ageing.[10 12] However, recent findings from cross-sectional studies suggest that this assumption is neither appropriate for all cognitive domains nor appropriate for all autistic adults. We, for example, found that while autistic adults reported that they are experiencing a large number of cognitive challenges in daily life, these autistic adults hardly differed from controls on a wide range of cognitive domains (eg, episodic verbal memory, inhibition[13–15]) and not all adults experience the same cognitive challenges.[15] The ASC group performed worse with respect to advanced social knowledge (ie, mentalising) and executive functioning (especially the ability to come up with various adequate solutions when confronted with a challenge, ie, generativity[15]). But, interestingly, the observed effect sizes were small and only a small proportion of the autistic adults seem to have clinically relevant cognitive problems. In studies including autistic children, typically medium effect sizes are observed and performance differences are apparent across a broader range of cognitive domains.[16] Moreover, independent empirical studies including old autistic individuals recently reported similar findings as Lever and colleagues observed.[17 18] So, in sharp contrast to the dominant hypothesis, these novel findings suggest that with increasing age, the cognitive differences between autistic adults and nonautistic adults become less pronounced. Yet, even more challenging for the dominant view, is that strong age-related cognitive differences are found in nonautistic adults, with respect to episodic visual memory and visual working memory, but *not* in autistic adults.[13 15] Also, while across different ages, autistic adults seemed to perform worse with respect to mentalising, performances of adults with and without an ASC diagnosis did not differ in those over 50 years.[15] Hence, recent findings concerning cognition across adulthood are not just in contrast with the accelerated ageing hypothesis but could even suggest that autistic adults are

less sensitive to the typical cognitive ageing effects. This is why we proposed an alternative hypothesis, which is that the cognitive strategies (a subgroup of) autistic adults use, while disadvantageous when young, are beneficial (ie, protective) when older.[12 15] However, in order to test such a hypothesis, we first need to determine whether previous cross-sectional findings can be replicated cross-sectionally as well as longitudinally, in sufficiently large groups. This is especially of importance as recently various small studies have been published with rather mixed findings regarding which cognitive functions are predominantly associated with specific cognitive ageing patterns.[18–22]

Moreover, we need to determine whether we can actually form well-validated adult ASC subtypes. If there are indeed adulthood subtypes, various ageing hypotheses (ie, accelerated cognitive decline, protective cognitive strategy use and delayed ageing) can be true for different subtypes in different cognitive domains. Ideally, when defining subtypes, one uses well-accepted low-cost, nonintrusive measures such as short questionnaires.[23] Behaviorally defined subgroups, so far, differed mainly in severity (ie, quantitative instead of qualitative differences[24]). However, it could well be that valid subtypes can be formed when symptom information is combined with other important behavioural characteristics, which are in the general population considered to be vulnerability (eg, stressful life events and lack of control) and protective factors (eg, physically activity and a sound social network) for a wide range of outcome measures.

These adulthood subtypes are not just of relevance for cognitive ageing but are also of relevance to gain more tailored information regarding someone's prognosis. Such newly formed subtypes need to be tested by checking multiple external validators, such as temporal stability and clinical outcome (next to change in cognitive abilities also the presence of comorbidities, subjective well-being) at follow-up. Ideally, in order to test the specificity of newly formed subtypes, another clinical group needs to be included. Put differently, including data from an additional neurodevelopmental condition (like attention deficit hyperactivity disorder (ADHD)) ensure that we can test whether the obtained subgroup findings will be specific for ASCs and follow the distinction between ASC and other neurodevelopmental conditions or whether the obtained subtypes cross the classical borders between these conditions. Given the overlap between ASC and ADHD,[25] the second scenario is more likely. Hence, the proposed study will not just gain relevant knowledge for those with an ASC diagnosis, but it will also add to the understanding of ageing in related neurodevelopmental conditions.

In sum, we have two major aims:

1. To arbitrate between contrasting ASC-related cognitive ageing hypotheses by studying ageing trajectories.
2. To determine (late) adulthood ASC subtypes and to explore whether there are specific prognostic behavioural markers that are related to the observed heterogeneity and can be used in clinical practice to

---

[i]We chose to use the term autism spectrum conditions as the traditional and official DSM-5 term 'autismspectrum disorder' and we use identity-first language as this is what a majority of autistic adults seem to preferin the UK. We are well aware that in other countries, like the Netherlands, the preference of autistic adults is notto use identity-first language, but given that this paper is written in English we chose the UK preference.

predict who might or might not be prone to (a) co-occurring mental health conditions, (b) lower quality of life and (c) cognitive decline.

We hypothesise that on overall group level, we will replicate the findings of our cross-sectional study,[13 15] cross-sectionally as well as longitudinally. This implies that for the majority of cognitive domains, the ageing trajectory of autistic adults will parallel the ageing trajectory of the controls, but for visual memory, visual working memory and theory of mind, the ageing pattern would be in line with the protective cognitive strategy hypothesis.

Moreover, we hypothesise that based on a subset of known vulnerability and protective factors for typical accelerated (cognitive) ageing next to ASC symptom measures, valid ASC adulthood outcome subtypes can be determined. However, currently there are no longitudinal data sets focusing on autistic adults, which include (a) a significant proportion of adults over 55 years of age, with (b) sufficiently detailed measures on mental health, quality of life and objective and subjective

cognition and with (c) comparison groups (COM) with and without other neurodevelopmental conditions (ie, ADHD). Each of the criteria needs to be met in order to properly test the aforementioned hypotheses. Moreover, most earlier studies did not report whether they consulted autistic adults in the research process while the potential outcomes of these study are especially relevant to them. Therefore, we designed the current accelerated longitudinal study with regular input from autistic adults.

Thus, in this study, we will map changes across time in cognition, mental health and subjective well-being in autistic adults and nonautistic adults in mid and late adulthood (30–90 years). This way we can test whether we can replicate earlier cognitive findings and we can test specificity, external validity, temporal stability, predictive validity and replicability for behaviorally defined subtypes in autistic adults.

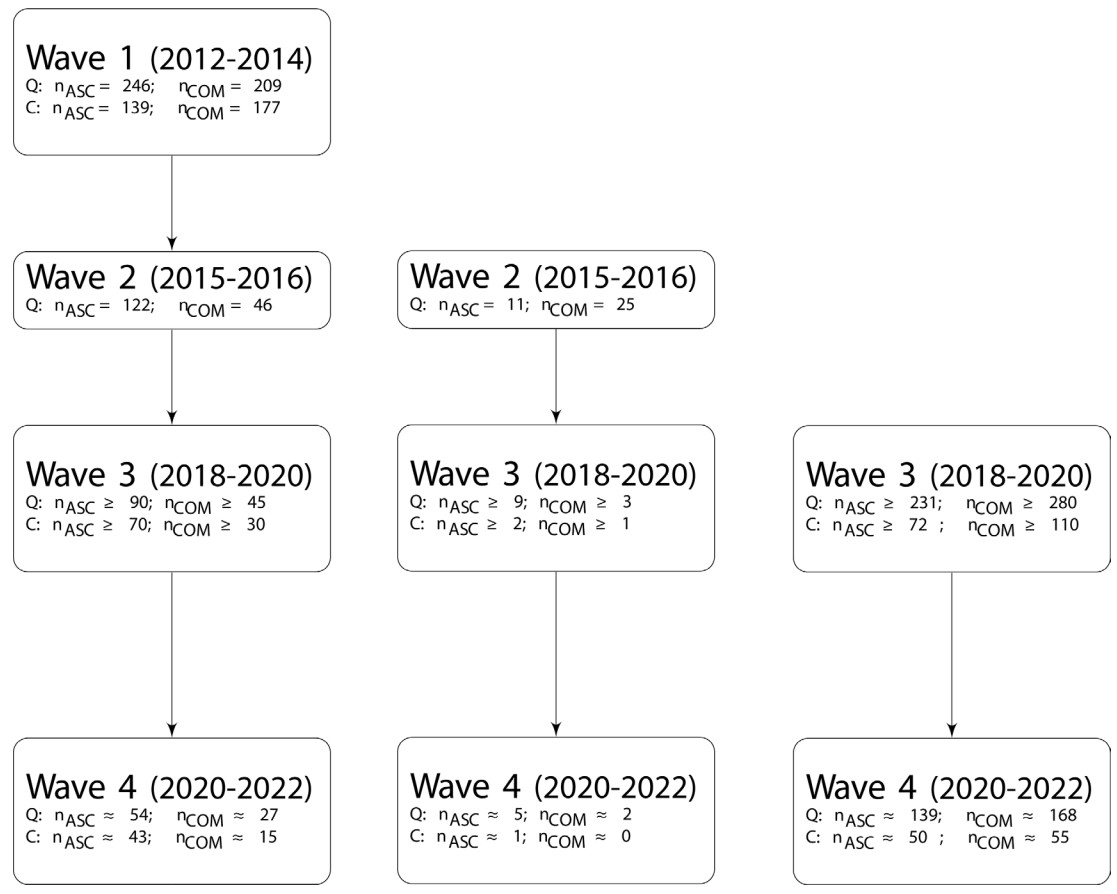

**Figure 1** Flow charts for each separate cohort. Cohort 1 (left panel) started in March 2012, cohort 2 (middle panel) started in December 2015. Cohort 3 started in September 2018. Data collection on wave 4 is estimated to start end 2020, numbers reported for wave 4 are based on expected attrition rates (ie, 40% for questionnaires, for cognition 30% for ASC, 50% for comparisons). Participant numbers in wave 3 are based on current inclusion in combination with our power analyses for wave 4. From wave 1 to wave 2, the inclusion age increased from 20 to 30 years old (see main text).=indicates exact numbers; ≥indicates planned numbers; ≈ estimated numbers based on expected attrition rates. Please note that we did not include information regarding the inclusion of the participants with an attention-deficit/hyperactivity disorder diagnosis as participants of one cohort will be, due to COVID-19, be recruited across waves 3 and 4. ASC, autism spectrum condition; C, cognitive tests; COM, comparison; Q, questionnaires.

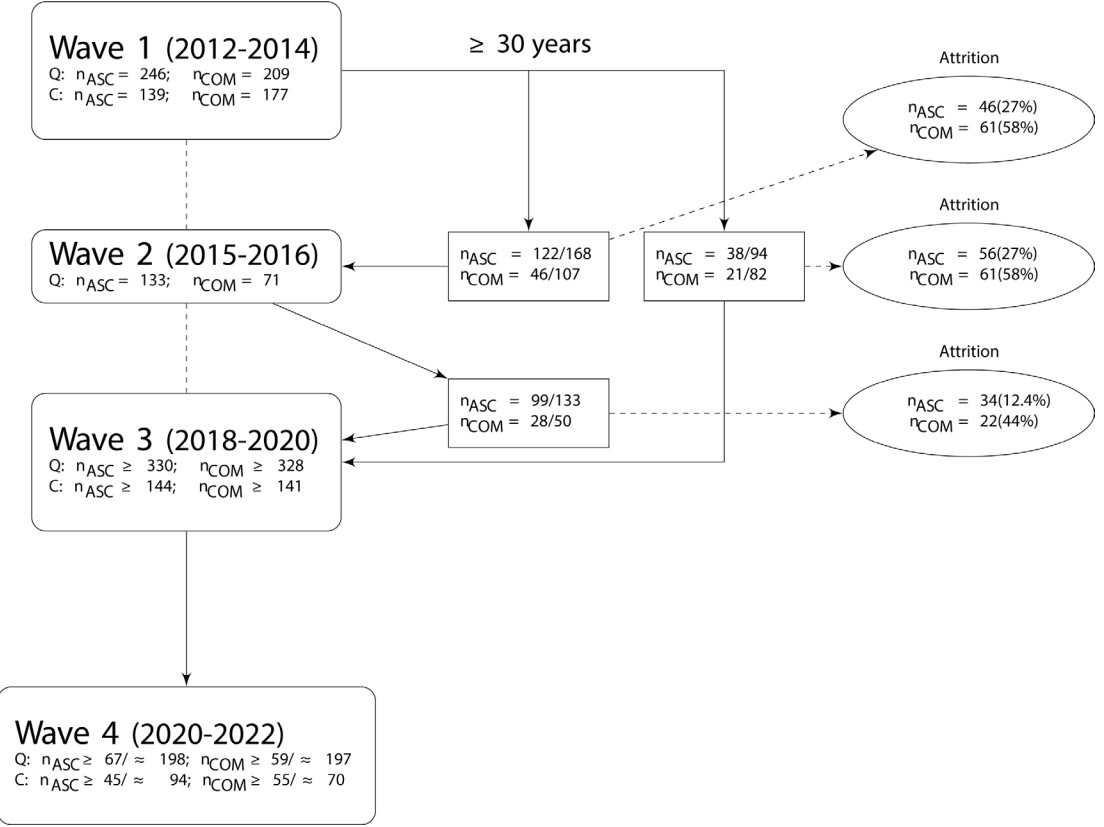

**Figure 2** Total number of participants seen (waves 1 and 2) or need to be seen minimally (waves 3 and 4) at each wave when taking into account the estimated attrition rate (see attrition) and age inclusion criterion (>30 years) at time of inclusion (see text). =indicates exact numbers;≥planned numbers; ≈ estimated numbers based on expected attrition rates. Please note that in this figure no information regarding the recruitment of participants with an attention-deficit/hyperactivity disorder diagnosis is provided. N.B. reported numbers at wave 4 are based on power analyses. The number of newly recruited participants can be seen in figure 1. ASC, autism spectrum condition; C, cognitive tests; COM, comparison; Q, questionnaires.

## METHODS

### Study design

This study follows a multistage overlapping cohort design assessing (cognitive) ageing in ASC including an ASC group, a typically developing COM and an ADHD group. This study was designed to obtain an accelerated longitudinal data set. Within each of these three cohorts, we start measuring at a different age point, but the cohorts do overlap in the older age ranges. Because of this, we will not have to follow each individual for a full lifetime, to be able to say something about development during adulthood. Essential is that within a relatively short time frame,[26] cognitive alterations can be detected from the age of 30–40 years onwards in healthy ageing, where some of the cognitive processes start to decline from 30 onwards (ie, processing speed), while others mainly decline after the age of 55 years (ie, verbal memory).[27] This protocol paper concerns four waves, which encompass three cohorts of people. Please note that in waves 1 and 2, we did not have funding yet for an accelerated longitudinal study. This means that these waves were originally set up as independent studies. The time frame of data collections is as follows: wave 1, March 2012 to July 2014 (cohort 1); wave 2, December 2015 to December 2016 (cohort 2); wave 3, September 2018 to ± October 2020 (cohort 3);

wave 4, November 2020 to ± December 2021. At the time of writing the current protocol, the COVID-19 pandemic happened. This did already affect the exact timing of the wave 3 and might also affect the timing of wave 4, but currently we do not know all exact consequences. One of the consequences that already occurred is that not all ADHD participants could be recruited during the time frame of wave 3, so approximately 50% of the participants of this specific group need to be recruited during wave 4. We, for now, consider these participants to be part of cohort 3 as this data will be collected close to the end date of wave 3 and, therefore, do not need to be considered a separate cohort. Please see figures 1 and 2 for a visual depiction of the different waves and cohorts.

### Procedure and setting

After potential participants showed interest in participation, they received a study information package. Only when we received informed consent, people were actually included in the study. Please note that participants of wave 1 were originally recruited for a cross-sectional study only, but the majority was asked to indicate whether we could contact them for future studies. Approximately 80% of the ASC group and approximately 40% of the COM group granted such permission. The wave 1 participants

between 30 and 80 years of age who gave an affirmative answer were partly recruited for wave 2, but again for this specific wave only. Of the newly recruited participants at wave 2, approximately 90% of the ASC group and approximately 65% of the COM group granted permission to contact them for future studies. At wave 3, participants were asked to participate in both waves 3 and 4, meaning that they needed to give consent to take part in two measurements at two distinct time points with a 1-year to 2-year interval. In each wave, participants received a series of questionnaires (on paper or via internet, see table 1 for details). The first selection (see inclusion/exclusion criterion) is based on the first series of questionnaires (eg, sociodemographics, diagnostic information, autism spectrum quotient (AQ), AD(H)D-SR)) participants filled out. If participants meet the inclusion criteria for the subtyping study, they fill out the second series of questionnaires. When based on the collected questionnaire

and interview data participants also meet the first series of inclusion criteria for the cognitive study (Wave 1, 3, and 4), they are invited for a face-to-face session at the university, or closer to the participant's home on request in order to reduce attrition rates as much as possible. In this first face-to-face session, the Autism Diagnostic Observation Scale (ADOS or ADOS-2; T1) or *Nederlands Interview ten behoeve van Diagnostiek Autismespectrumstoornis bij volwassenen* (Dutch Interview for assessment of autism spectrum disorders in adults; NIDA; T2), the International Neuropsychiatric Interview (MINI), mini mental state examination (MMSE) and Wechsler Adult Intelligence Scale (WAIS) subtests are administered when it concerned autistic adults, but the ADOS(-2) (or NIDA) is not administered in the COM groups (see table 2). In waves 3 and 4, a questionnaire called the menopause rating scale (MRS) is added to this first session in a subset of the participants. If the autistic participants still meet inclusion criteria, a

**Table 1** Timing and use of questionnaires

| Measures | | Waves | | | | Study focus | |
|---|---|---|---|---|---|---|---|
| Instrument | Construct | 1 | 2 | 3 | 4 | Aim 1 | Aim 2* |
| General | Descriptives | � | | | | INCL | INCL/SUB |
| AQ- self | Autism traits | | | | | INCL | SUB |
| AQ- proxy | Autism traits | | | | | | |
| IRI-self | Personal relations | | | | | | |
| IRI-proxy | Personal relations | | | | | | |
| SSQ-self | Sensory sensitivities | | | | | | SUB |
| SSQ-proxy | Sensory sensitivities | | | | | | |
| ADHD-SR | ADHD traits | | | | | INCL | |
| SCL-90 | Mental health | | | | | | VAL |
| Health Ques | Physical health | | | | | INCL | |
| MRS | Menopause | | | | | | |
| CFQ | Global cognition | | | | | REP | VAL |
| PANAS | Global emotion | | | | | | SUB |
| WHOQoL | Quality of life | | | | | | VAL |
| Happiness | Well-being | | | | | | |
| Mastery | Stress/self-control | | | | | | SUB |
| Worry | Stress/worries | | | | | | SUB |
| IPAQ | Physical activity | | | | | | SUB |
| Brugha | Life events | | | | | | SUB |
| CPI | Social network | | | | | | SUB |
| CAT-Q | Camouflaging | | | | | | |

Aim 1: arbitrate between cognitive ageing hypotheses; aim 2: subgrouping; Grey shading indicates that a measure was administered during that wave.

*Please note that more than one variable can be derived from the instruments. For example, we will use 14 cluster variables for our subtyping analysis, which are based on eight different instruments.

ADHD-SR, attention deficit hyperactivity disorder self report; AQ, autism spectrum quotient; Brugha, Brugha questionnaire of important life events during childhood and in the past years; CFQ, cognitive failure questionnaire; CPQ, close persons questionnaire to measure present social support from the four most intimate persons; Happiness, perceived well-being question of the NAR; INCL, used for inclusion; IPAQ, international physical activities questionnaire; IRI, interpersonal reactivity inventory; MRS, menopause rating scale; PANAS, positive and negative affect schedule; REP, replication; SCL-90, symptom checklist; SSQ, sensory sensitivity questionnaire; SUB, used for subtyping; VAL, validation; WHOQoL, WHO health organisation quality of life.

**Table 2** Timing and use of diagnostic measures

**Diagnostic assessment**

| Type | Instrument | Construct | Waves 1 | 2 | 3 | 4 | Study focus Aim 1 | Aim 2 |
|------|-----------|-----------|:---:|:---:|:---:|:---:|:---:|:---:|
| CNP | MMSE | Global cognition | ▪ | | ▪ | | INCL | |
| CNP | WAIS subtests | Estimation of intelligence | ▪ | | ▪ | | INCL | INCL |
| OBS | ADOS | ASC | ▪ | | ▪ | | INCL | |
| INT | NIDA | ASC | ▪ | | ▪ | | | |
| INT | MINI | Mental health | ▪ | | ▪ | | INCL | VAL |

Grey shading indicates that a measure was administered during that wave.
ADOS, autism diagnostic observation scale; CNP, clinical neuropsychologic task; INCL, used for inclusion; INT, interview; MINI, MINI the International neuropsychiatric interview; MMSE, mini mental state examination; NIDA, Nederlands interview ten behoeve van Diagnostiek Autismespectrumstoornis bij volwassenen ((Dutch Interview for ASD assessment in adults)); OBS, observation; VAL, measure to test validity of subtypes.

second face-to-face session will take place in which the cognitive tests will be administered (see for the specific tests, table 3). For the nonautistic/non-ADHD comparison (COM) group, the two face-to-face sessions were combined on request. This implies that in theory in this group it could be that subgroup of COM participants did not meet the inclusion criteria for the cognitive section of the study but still did do the cognitive tasks. Due to the COVID-19 pandemic, the first face-to-face session was transferred to an online setting, but for the ADOS and for the cognitive testing, such a transition is not possible, so these sessions remain face-to-face. In each wave where cognitive testing is part of the protocol, there were 16 predetermined test orders that are counterbalanced across participants. Both sessions took place in a quiet distraction-free room in one or two sittings (each approximately 2 hours). In wave 1, control participants received a small additional fee for participating. In waves 3 and 4, all participants will also receive additional financial compensation for filling out the questionnaires as well as the face-to-face sessions (maximum 30€). In all waves, travel costs are reimbursed. Some participants of wave 1 were also invited to take part in an MRI study,[28–30] but that study is not included in this protocol.

### Study population and recruitment

For cohort 1, individuals between 18 and 80 years were recruited (recruitment wave 1), for cohort 2,

**Table 3** Timing and use of cognitive measures

**Cognitive assessment**

| Type | Instrument | Construct | Waves 1 | 2 | 3 | 4 |
|------|-----------|-----------|:---:|:---:|:---:|:---:|
| CNP | RAVLT | Episodic verbal memory | ▪ | | ▪ | ▪ |
| CNP | WMS | Episodic visual memory | ▪ | | ▪ | ▪ |
| CNP | DAT and COWAT | EF (generativity) | ▪ | | ▪ | ▪ |
| CNP | TMT A and B | EF (flexibility) | ▪ | | ▪ | ▪ |
| CNP | ToL | EF (planning) | ▪ | | ▪ | ▪ |
| EXP | Go-NoGo | EF (inhibition) | ▪ | | | |
| EXP | Simon | EF (inhibition) | ▪ | | | |
| EXP | N-back | EF (working memory) | ▪ | | | |
| EXP | 2-Choice RT | Processing speed | ▪ | | | |
| CNP | Faux Pas | Social cognition (ToM) | ▪ | | | |
| EXP | Odd ball | Attention | ▪ | | | |
| EXP | ABT | Prospective memory | | | ▪ | ▪ |

Grey shading indicates that a measure was administered during that wave.
ABT, Amsterdam breakfast task; 2-Choice RT, a simple computerised reaction time task; CNP, clinical neuropsychologic task; DAT, Dutch version of the controlled word association test; EF, executive functioning; EXP, experimental task; RAVLT, rey auditory verbal learning test; Simon, a computerised inhibition task; ToL, tower of London; ToM, theory of mind; WMS-III, Wechsler memory scale third edition, subtest visual reproduction.

individuals over 45 years were recruited (recruitment wave 2), for cohort 3, individuals between 30 and 90 years were recruited (recruitment wave 3). Individuals with ASC and/or ADHD were recruited through several mental health institutions across the Netherlands, by means of advertisement on client organisation websites and newsletters, and through social media (ie, Twitter and LinkedIn). Comparison participants were recruited through social media and the personal networks of the researchers and participants. In order to ensure an equal proportion of participants across the whole age range and in order to match the COM with the ASC group, people are recruited within age bins of 10 years of age and with a sex distribution of 2:1 (men:women).

Inclusion criteria for all participants were (1) no intellectual disability, as reported on questionnaires or an IQ>70 as estimated using two subtests of the WAIS III or IV-NL (WAIS-III[31]; WAIS-IV-NL[32 33]) and (2) sufficient understanding of Dutch language (ie, at least one Dutch parent, Dutch was spoken in their family or when participants fill in more than 90% of the questionnaires).

An additional inclusion criterion for the ASC or ADHD group was a prior clinical ASD and/or ADHD diagnosis according to the DSM-4 or DSM-5.[1 34] For the COM group, the exclusion criteria were: (1) clinical diagnosis of ASC or ADHD in themselves or direct family members, (2) an elevated score on the AQ>32 (AQ[35]; or an elevated score on ADHD-self report (ADHD-SR)≥6[36] and/or (3) reporting more than one psychosis.

For cognitive testing, additional inclusion criteria were applied. All participants have to have (1) MMSE >18[37], (2) no history of neurological disorders (eg, epilepsy, stroke, cerebral contusion), (3) no current alcohol or drugs dependency and (4) no more than one psychosis.

## Measures

All measures and the waves at which they were administered are depicted in tables 1–3.

## Questionnaires

First, a series of general questions were asked regarding, among others, birth year and month, gender, education, diagnostic history, ASC/ADHD in family members and co-occurring conditions. Second, standard questionnaires are included, which assessed several domains of functioning, namely: (a) ASC domains including general autism traits (AQ[35 38]), interpersonal reactivity[39 40], sensory sensitivities[15 41], (b) mental health including ADHD characteristics (ADHD-SR[36]) and general mental health problems (symptom checklist-90[42 43]), (c) physical health (Health questionnaire[44]) and physical activity (international physical activities questionnaire[45]), (d) well-being (happiness[46]) and quality of life (WHO health organisation quality of life-bref[47 48], (e) stress and worries (Worry Scale/Fear Questionnaire[49 50]) and stress and self-control (Pearlin Mastery Scale[51]), (f) subjective cognitive complaints ([52 53]cognitive failure questionnaire), (g) important life events (Brugha[54 55]), (h) emotions (positive and negative affect schedule[56 57]), (i) social network (Close Persons Questionnaire [58]), (j) camouflaging of autistic traits [54 59], (k) menopausal complaints (MRS[60]). The measures were chosen based on the following criteria: (a) valid to be used in ageing and/or autism-related studies, (b) easy to administer and (c) inexpensive to administer. Moreover, an additional inclusion criterion for the measures specifically included for aim 2 is that these instruments measure factors that have predictive value for ageing-related outcome measures of interest for autistic adults (ie, cognition, psychological complaints and well-being).

## Diagnostic assessment

Global cognitive functioning was assessed with the MMSE[37 61]; and general intelligence was estimated by two subtests (Vocabulary and Matrix Reasoning) of the WAIS-III wave 1/WAIS-IV waves 3 and 4[31–33]. Autistic traits were assessed with either the ADOS[62 63] or the NIDA,[64] dependent on the measurement, the participant was performing (ie, ADOS was administered at the first measurement moment (T1) for each of the autistic participants and the NIDA at subsequent moments (T2). Psychiatric comorbidity was assessed with the MINI[65 66] in all participants independent of group.

## Cognitive assessment

Participants completed a comprehensive battery of cognitive assessments in the following domains: (a) memory, including episodic verbal memory (Rey auditory verbal learning task[67]), episodic visual memory (Wechsler memory test-III[68]), prospective memory (Amsterdam breakfast task[69 70]; in-house development author APG), working memory (N-Back[13]), (b) social cognition (Faux pas; theory of mind[71 72]), (c) processing speed (2-choice reaction time (2-CRT)[15], (d) executive functioning including generativity (Dutch version (DAT) of the controlled word association test[73]), planning (Tower of London[74]), inhibition (Simon task,[14] Go/Nogo task; (in-house development by AGL), flexibility (trail making test A&B[75]), (d) attention (oddball; in-house development by AGL).

## Data management and quality control

Following the institutional and national guidelines, the data were pseudonymised and encrypted during and after data collection. For quality control, the data of all Waves were entered by participating master students and research assistants and between 10% and 100% was checked by other master students, research assistants and the PhD students involved in waves 1, 3 and 4. The initial percentage checked depended on who entered the data for the first time and whether mistakes were discovered. When mistakes were discovered in the first checked subset, there was a 100% check.

## Patient and public involvement

Before the start of wave 3, we formalised the involvement of the major stakeholder group, the autistic adults. We

**Table 4**  Contributions to the protocol

|  | HMG | AGL* | IG* | TR | JAR | CT | WvdP | APG |
|---|---|---|---|---|---|---|---|---|
| Funding | X |  |  |  |  |  |  |  |
| Study design wave 1 | X | X |  |  |  |  |  |  |
| Study design wave 2 | X |  |  |  |  |  |  |  |
| Study design subgrouping (across waves) | X |  |  | X | X |  |  |  |
| Study design cognition (across waves) | X |  |  |  |  | X |  | X |
| Data collection |  | X | X | X | X | X | X | X |
| Set up/feedback statistical analyses | X |  |  | X | X | X | X | X |
| First draft of the manuscript | X |  |  |  |  |  |  | X |
| Feedback on subsequent versions of manuscript | X |  |  | X | X | X | X | X |
| Final approval of the manuscript | X |  |  | X | X | X | X | X |

*Were not involved in preparation of the current manuscript and design of the longitudinal study, but were involved in earlier stages of the research, see also the Acknowledgements.

AGL, Anne Geeke Lever; APG, Annabeth P. Groenman; CT, Carolien Torenvliet; HMG, Hilde M. Geurts; IG, Iuno Groot; JAR, Joost Agelink van Rentergem; TR, Tulsi Radhoe; WvdP, Wikke van der Putten.

meet a group of four older autistic adults at least three times a year for a 4 year period in which we discuss, for example, study designs, questionnaires, letters, tasks and task instructions, study results and the dissemination of these results. They are paid for their contribution. In earlier waves, the involvement of the stakeholders was not formally organised.

### Data analysis plan and power

The analyses of wave 1 are described in detail in the published papers.[13–15 76 77] Data analysis plan related to predominantly wave 3 data is registered on As Predicted. For example, a detailed statistical analysis plan is provided for our cross-sectional replication of our previous cognition findings at https://aspredicted.org/blind.php?x=at2iq2. We also provided a detailed analysis plan for subgrouping on T1 wave 2 and wave 3 data by means of community detection at https://aspredicted.org/blind.php?x=hu4ey6. The protocol authors (table 4) will preregister final and detailed analysis plans regarding the combined data across the differences waves on AsPredicted or similar platforms before data collection of wave 4 ends. In these plans, we will also include attrition rate handling, outlier detection, and how we will deal with item-level, instrument-level or time-point-level missing data depending on the question at hand.

In order to determine whether we can replicate the cross-sectional findings with an accelerated longitudinal dataset (ie, aim 1), we will, for example, apply dynamic growth models, to fit the developmental (ageing) shape of each cognitive process. The proposed set-up ensures that individual data points can be plotted against time (for an ASC example[78]) and that we can actually build a linear mixed model of change in which we can model individual change, potential cohort effects and potential measurement effects. Moreover, these techniques can provide information on whether growth curves differ shape for different cognitive subdomains. For these analyses, we primarily will use data points for those participants of whom we have cognitive data at two timepoints (T1 and T2). For aim 2, we will mainly use a bottom-up approach, where community detection will be used in order to form subgroups for which we perform an independent replication, focus on external validation on T1, longitudinal stability of subgrouping and predictive validity from T1 to T2. While we predict that these T1-based subtypes will differ in, for example, their cognitive profiles, comorbidities and subjective well-being at both T1 and T2, there is no guarantee that these subtypes will be the subtypes that actually differ in their cognitive ageing process. Therefore, next to the aforementioned approach, we will use a second top-down approach in which the cognitive change will be taken as the starting point. For this second approach, we did not power the study and this approach is more explorative in nature, but in this way, we can focus on the heterogeneity regarding cognitive ageing. First, we will determine the amount of cognitive decline one shows. This will be determined in the subset of participants for which we have cognitive data by using multivariate normative comparison (MNC) to quantify the severity of cognitive impairment. MNC can provide, next to a dichotomous measure, a continuous measure that reflects the degree of cognitive deviation. Next, we will explore which measures predict whether or not someone will show a cognitive decline, stays the same or has an increased cognitive performance.

In order to detect a (continuous) change with the planned statistical methods on the included cognitive measures (aim 1), we will need a sample size of ±100 participants in the ASC group and 85 participants in the COM group at two time points when assuming an effect size of 0.20 and power of 0.80.[26 78] In order to reach at least these numbers, we will need to recruit more T1 participants than needed for these analyses to (a) be able to match on age and sex and (b) to assure sufficient participants at

T2 when considering attrition rates of 30% (ASC group) to 50% (COM group). The inclusion criteria (see earlier) are the strictest for the cognitive study, which implies we need to oversample when collecting the questionnaire data as participants needed for aim 1, will be a subset of the participants needed for aim 2. Required sample sizes for aims 1 and 2 differ. For aim 2 (subtyping), it is of importance that we include a sufficiently large number of participants to have (a) a training data set to determine subgroups based on 14 variables of 8 different questionnaires and (b) a test data set to determine whether the subgroup solution based on the same 14 variables is similar to the training data set. This data need to be collected in both wave 2 (training data set) and wave 3 (test data set) combined. Moreover, to test the temporal stability of the subgroups obtained, we require T2 data from different cohorts of participants (waves 3 and 4). We expect subgroups both within the ASC group and in a combined sample of ASC+COM. Through simulation, we found that to reliably recover three subgroups, we require at least 160 participants. Therefore, we need to collect data from a minimum of 240 participants at T2 (160 ASC, 80 COM, see also figures 1 and 2). In addition, we collect a minimum of 30 participants with ADHD at T2. The attrition rate from wave 3 to wave 4 is expected to be lower (=/- 40% across ASC and COM groups) for the questionnaire data given that people do not need to come to the lab. For the ADHD group, we estimated the attrition rate to be ±50%. Please note that we will compare those individuals who were successfully followed up to those lost to follow-up on the following variables: Autism and ADHD severity (based on AQ and ADHD-SR), age, estimated IQ and the presence comorbid conditions (both psychiatric and somatic) in order to examine the impact of attrition on our findings.

## Ethics and dissemination

This study is low risk in terms of ethical considerations, since participants will not receive an intervention or rules to live by. Moreover, there is no medical risk associated with participation. Apart from time and energy investment, there is no 'burden' associated with participation. We chose to exclude those questions in the interview that were meant to determine whether there are current suicidal thoughts as affirmative answers as such questions could give raise to an ethical dilemma. Moreover, while we do use questionnaires and interviews which are often used in clinical practice during ASC assessment, these instruments cannot be used as standalone diagnostic instruments. Hence, based on these instruments, one cannot conclude for clinical purposes whether someone received an unjustified ASC diagnosis or whether they should have an additional diagnosis of a co-occurring condition. Consequently, we did not communicate about the individual outcomes with participants of these tests in accordance with the rules as set by the local ethical review board. Ethical approval for this study was obtained from the local ethical review board of the department of Psychology of the University of Amsterdam (wave 1, 2011-PN-1952 and 2013-PN-2668; wave 2, 2015-BC-4270; waves 3 and 4 2018-BC-9285, see also online supplemental material 1. Local research committees of the participating clinical centres will often need to give additional approval before one can recruit participants at this specific centre but will not need to conduct an additional ethical review given that the ethical approval is already covered by the ethical review board of our university. Thus, all procedures performed were in accordance with the ethical standards of the institutional ethical and research committees and with the 1964 Helsinki declaration and its later amendments or comparable ethical standards.

Study results will be published in peer-reviewed (inter) national journals (open access) and presented at conferences. Participants of all waves will receive information regarding the outcomes on a group level and receive a newsletter throughout the duration of the study when they consented to receive such information. Furthermore, these yearly newsletters are also published on our website (dutcharc.nl) and at the end of this overarching project, a conference will be organised to which different stakeholders (participants, autistic adults and their relatives, clinicians and researchers) will be invited. In both 2016 (main focus participants and clinicians of participating clinical centres) and 2018 (all aforementioned groups) such conferences were already successfully organised.

**Acknowledgements** We want to thank the past, current, and future participants. The students who helped with the data collections as previous, current, and future research assistants. We also thank the institutions and organisations who helped/are helping with recruitment. We are especially grateful by the time and advice provided by the older autistic adults in our think tank. We mention two people by name Anne Geeke Lever (AGL) and Iuno Groot (IG) as AGL was as first PhD student on a related project instrumental for the set up and data collection of Wave 1 and IG, as research assistant, for the data collection of Wave 2. Therefore, we thank both AGL and IG for their involvement regarding the first studies related to autism and ageing on which we build the future measurements waves.

**Contributors** All authors declare that they contributed to the content of this manuscript and commented on subsequent versions of the manuscript and were involved in parts of the design of the overall study. Each of the authors comments on each other's statistical analysis plans before it is submitted to AsPredicted/Open Science Framework. Please see table 4 for an easy overview of the contributors of each of the authors and of people who were involved in an earlier stage (see also Acknowledgements). In short, HMG secured funding, was responsible for the overall design related to each of the waves and regarding the initial set up for both the subtyping as well as cognition study across waves, is involved in the set up for the statistical analyses, and wrote, together with APG, the first draft of this manuscript. JAvR and TR were involved in the study design and statistical analysis plan of the subgrouping study. CT and APG were involved in the study design and statistical analysis plan of the cognition study. TR, CT and WJvdP are collecting the data in waves 3 and 4 and, together with HMG and an external collaborator, involved in the statistical analysis plans for all questions related to camouflaging. TR, CT and WJvdP are double checking all data, and APG and HMG are supervising CT, JAvR and HMG are supervising TR, and HMG is supervising WvdP together with an external collaborator who is not involved in the setup of the current study or in the data collection.

**Funding** This study was funded by the Grant Sponsor is the Netherlands Organization for Scientific Research (NWO); VIDI Grant (grant number: 452–10-003 and VICI Grant (grant number: 453–16-006) awarded to HMG. This funding organisation had no involvement in study design, analysis or interpretation of the data but does require open access publication of the findings. Both project

grants were reviewed by three (VIDI) to five (VICI) anonymous and independent researchers.

**Competing interests**  None declared.

**Patient consent for publication**  Not required.

**Provenance and peer review**  Not commissioned; externally peer reviewed.

**ORCID iDs**
Hilde M Geurts http://orcid.org/0000-0002-4824-9660
Joost A Agelink van Rentergem http://orcid.org/0000-0002-1600-8635
Tulsi Radhoe http://orcid.org/0000-0001-9271-9503
Carolien Torenvliet http://orcid.org/0000-0002-0865-5154
Wikke J Van der Putten http://orcid.org/0000-0001-5411-2831
Annabeth P Groenman http://orcid.org/0000-0002-8394-6605

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
