## [Reviewer comments · BMJ Open]

ARTICLE DETAILS

TITLE (PROVISIONAL)	Aging and heterogeneity regarding autism spectrum conditions: A protocol paper of an accelerated longitudinal study
AUTHORS	Geurts, Hilde M.; Agelink van Rentergem, Joost; Radhoe, Tuls; Torenvliet, Carolien; Van der Putten, Wikke J; Groenman, Annabeth P

VERSION 1 – REVIEW

REVIEWER	Dario Siniscalco University of Campania, Naples, Italy
REVIEW RETURNED	13-Jul-2020

GENERAL COMMENTS	My only concern regards the title. The authors used the word "senescence", but this term is not more used in the text. In addition, it is usually a term designed for molecular mechanisms of aging cells. I suggest to use the term "aging".
---

REVIEWER	Eric Rubenstein Boston University, United States of America
REVIEW RETURNED	01-Sep-2020

GENERAL COMMENTS	I commend the authors for receiving funding and doing work on the crucial topic of aging in autism. It is great to see the design being registered and I am very excited to see the results of this study in coming years. I have a few areas where I think more clarification is needed and I think there is some nuance that can be added to the introduction, as there is a reason why we do not have much knowledge about autism in adults. 1. There is very little detail on the analytic plan. You mention that analyses from Wave 1 are in 4 published papers. Are you planning on repeating those? How are you adapting to longitudinal data? You note that you have registered the plans on AsPredicted, which is great, but there is no link to that site. I think more information on the analysis is needed because the design is complicated. a. Cohorts 1 has 4 time points; Cohort 2 has 3 time points and Cohort 3 has two time points. How is that accounted for in your analysis? b. It is unclear how you will create the trajectories- latent trajectory analysis? Will you include all measures in it? What are the 14 variables you chose and why did you choose them? It seems like there are 8 variables listed in table 1. c. How will you handle missing data? d. Will you have power to look at different age groups? You match to control by age but if you wanted to see trajectories of 55+ are you powered to do that?
---

	2. So the aim 1 and 2 samples are different with aim 1 being a subset of aim 2? If that is correct, then you can't use cognition variables in your trajectory analysis, since some won't have those measurements? It would seem like they would be important variables to include, especially considering the epilepsy is so common. 3. I found the figures more confusing than helpful. Can you add in calendar time to aid the reader in understanding the course of this project? 4. It would be good to add how you will examine whether attrition is differential and how you will correct for that if it is the case. For example, if the people that don't show up for the visit don't show up because their health is so poor then that will bias your results. 5. It is not entirely clear how this is an accelerated longitudinal design. Wouldn't you want to be recruiting more distinct age groups in each cohort? It seems like everyone after wave 1 would be 30-90, which is a very big range and will likely miss big gaps of the age range. 6. Similarly, what are you expecting in terms of your age distribution and gender distribution? 7. Outside of analysis, the paper needs a discussion of why there have been so few studies of older autistic adults and how that impacts this study. Simply, we are not good at identifying autistic adults as having autism. There is no gold standard measure. Further, there are major cohort effects in autism, as someone that was diagnosed in 2016 at 6 years old will have a world that is very different towards autism as compared to someone diagnosed in 2016 at 60. I think that there needs to be some acknowledgement of all the external factors that impact a person on the spectrum and how those may drive cognitive trajectories or subgroups and how that may be confounded by cohort (as in age-period-cohort analyses). What does it mean that you are excluding those without a past ASD diagnosis? Does that exclusion make them more or less similar to controls? 8. Subgroups in autism tend not to replicate. Consider tempering the paragraph in the conclusion about using the subgroups as they will need to be replicated using external data before clinical changes should take place. 9. The word senescence does not appear anywhere but the title. I had to look up what it meant after I read the paper because I was confused not seeing it in the text.
--	---

REVIEWER	Stephen M. Edelson Autism Research Institute United States of America
REVIEW RETURNED	05-Nov-2020

GENERAL COMMENTS	The manuscript titled "Senescence and heterogeneity regarding autism spectrum conditions: A protocol paper of an accelerated longitudinal study" [bmjopen-2020-040943] describes a much-needed research project that will likely provide much insight about adults on the autism spectrum as they age into their senior years. I have listed my comments below: The authors discuss two aging hypotheses, i.e., accelerated cognitive decline and protective cognitive strategy use. Another
---

	possible hypothesis often reported in those who are developmentally delayed or mentally challenged is a gradual maturation (i.e., improvement) in cognitive function over time. It is not clear whether early developmental events will also be collected. Some critical events will likely be well-remembered, such as pregnancy and her birth complications, age began walking and talking, and the presence of seizures. Will there be any attempt to document, in relation to age, whether the participants received intensive behavior therapy, sensory/OT interventions, medications, and so on? It is quite possible that certain therapies, especially administered during childhood, could determine a range of developmental trajectories in relation to each subtype. I suggest that the authors consider administering a questionnaire designed specifically for sleep disturbances associated with autism. Sleep is quite prevalent among those on the spectrum, and quality of sleep may impact cognition (e.g., inattention, memory retrieval) and behavior (irritability, aggression). Page 12 lines 26 and 36. The authors refer to the COM group. What does this refer to? Earlier they use the term COMP to refer to a comparison group (page 10, line 24). Will there be any discussion with the participant and/or family members regarding postmortem brain research? I commend the authors' efforts in undertaking such an exciting and important research project.
--	---

VERSION 1 – AUTHOR RESPONSE

Reviewer Comments to Author:

Reviewer: 1

Reviewer Name: Dario Siniscalco

1.1. My only concern regards the title. The authors used the word "senescence", but this term is not more used in the text. In addition, it is usually a term designed for molecular mechanisms of aging cells. I suggest to use the term "aging".

Re: Thanks for this comment. This term was used in the original title of the approved grant proposal, so we thought it would be good to be consistent. However, while senescence is a developmental term comparable to adolescence (just a different life phase) it is not commonly used and we did indeed not use it ourselves in our main text. Therefore, we excluded this from the title and just refer to aging.

Reviewer: 2

Reviewer Name: Eric Rubenstein

I commend the authors for receiving funding and doing work on the crucial topic of aging in autism. It is great to see the design being registered and I am very excited to see the results of this study in coming years. I have a few areas where I think more clarification is needed and I think there is some nuance that can be added to the introduction, as there is a reason why we do not have much knowledge about autism in adults.

Re: We thank the reviewer for acknowledging the importance of our work. Our reply to your concerns is voiced in a point-by-point manner below.

2.1. There is very little detail on the analytic plan. You mention that analyses from Wave 1 are in 4 published papers. Are you planning on repeating those? How are you adapting to longitudinal data? You note that you have registered the plans on AsPredicted, which is great, but there is no link to that site. I think more information on the analysis is needed because the design is complicated.

Re: We agree that we did not include much detail as we felt we did not have sufficient space. This is why we referred to the fact we will submit and have submitted preregistrations. However, all your questions are valid and important and as the conclusion section could be excluded (see #0.1) we are pleased that in the revised manuscript we could add some more detail.

In our analyses, we will focus on replication in Wave 3 data of previous findings of Wave 1 data (i.e., running similar analyses in similar cross-sectional data, see for example <https://aspredicted.org/blind.php?x=at2iq2>, now also mentioned on page 15 of the revised manuscript) as well as on new analyses (i.e., running specific analyses given the specific type of data and the questions at hand, see for example subgroup analyses <https://aspredicted.org/blind.php?x=hu4ey6> now also mentioned on page 15 of the revised manuscript and for an example of a totally different research question see here: <https://aspredicted.org/blind.php?x=3ts7kv>).

Please note that we did not preregister all specific aspects of the project yet, as this will be part of the PhD trajectory of the PhD students for whom an important part of their trajectory and learning process is to determine all details in a statistical analysis plan and/or form specific hypotheses based on the most recent literature. Moreover, we want to make use of the most recent and sophisticated statistical analyses, and since this is a longitudinal study, registering all plans at the beginning of Wave 3 would have made this impossible. This implies that the preregistration of the analyses of some specific aspects of this data will be done during data collection in Wave 4.

As describing all these different specific analyses is beyond the scope of this protocol paper, we did give examples of planned analyses, and added some more details to address various aspects of the important questions raised by this reviewer. We especially focus briefly on how we handle the longitudinal data of different cohorts with different time intervals between the measurements. We also highlight how Aim 1 and Aim 2 are combined in specific analyses as we will employ two different methods of forming subgroups (Aim 2, see page 15 & 16 of the revised manuscript).

As the data analysis plan section is extended considerably, we did not copy and paste it in here, but refer to this section in the revised manuscript. We hope this does not cause any inconvenience.

a. Cohorts 1 has 4 time points; Cohort 2 has 3 time points and Cohort 3 has two time points. How is that accounted for in your analysis?

Re: We will account for these different time points differently depending on the question at hand, as some of the measures have changed across these different cohorts. So only for some measures and for a relatively small sample we do have 3 or 4 time points, but for the majority of

measures we will have 2 time points for a large part of the sample. In our main longitudinal analysis, we will focus on this specific part of the sample. Thus, for some of our questions we collapse data across cohorts (for example see <https://aspredicted.org/blind.php?x=hu4ey6>), and for some of our questions we focus first solely on those individuals for whom we have exactly the same measures on at least two time points.

In reality the situation is a little more complicated, with many participants participating in just one or two time points (just in Wave 3, in Wave 1 and 3, in Wave 3 and 4, etc.). To make optimal use of all the data, we can use linear mixed effects modeling for those questions where we focus on change across time to estimate fixed and random effects while using all data across all Waves and measurements. This allows us to estimate how much variance is due to cohort differences, measurement occasion differences, and individual differences. Furthermore, because these longitudinal data essentially constitute a missing data problem, linear mixed effects modeling is required for its ability to flexibly estimate models with Full Information Maximum Likelihood.

Nonetheless, as aforementioned, we will first have a less complex approach by focusing mainly on change from T1 to T2 (see also page 15 of the revised manuscript, and #2.2a)

b. It is unclear how you will create the trajectories- latent trajectory analysis? Will you include all measures in it? What are the 14 variables you chose and why did you choose them? It seems like there are 8 variables listed in table 1.

Re: We intend to use a linear mixed effects model for our latent trajectory analysis on the accelerated longitudinal data, to estimate all types of fixed and random effects (please also see comment #2.2a). We are interested in knowing whether the change over time (for example in cognition) is related to initial differences, but with different participants starting at different time points and participating in different time points, we have to choose a flexible statistical approach.

We will not include all variables in this latent trajectory analysis of change over time, as they are not all of primary interest. Of primary interest are the cognitive measures, measured in wave 1, 3, and 4, and a series of specific questionnaires administered in of wave 2, 3, and 4.

Based on 8 questionnaires, we include 14 variables as from some questionnaires more than one relevant dependent measure is derived. The 8 questionnaires listed for subtyping analyses were chosen based on a) whether or not it is a measure of a known factor which impacts both aging in relation to cognition and in relation to mental health and well-being (please see page 13 of the revised manuscripts; and b) whether we measured this in Wave 2 as this data will be collapsed with T1 data of Wave 3 (see <https://aspredicted.org/blind.php?x=hu4ey6>).

Page 16: "14 variables of 8 different questionnaires"

Page 13: "The measures were chosen based on the following criteria: a) valid to be used in aging and/or autism related studies; b) easy to administer; and c) inexpensive to administer. Moreover, an additional inclusion criterion for the measures specifically included for Aim 2 is that these instruments measure factors which have predictive value for aging related outcome measures of interest for autistic adults (i.e., cognition, psychological complaints, and well-being). "

Page 28: "Please note that more than one variable can be derived from the instruments. For example, we will use 14 cluster variables for our subtyping analysis which are based on 8 different instruments."

c. How will you handle missing data?

Re: One can distinguish different types of missing data. Here we can distinguish item-level, instrument-level, or time-point-level missing data. On the item-level, we have a rule that up to 10%

missing responses on an instrument can be imputed; else there are too many missing responses and we will consider the score missing on instrument level. On the instrument-level, it depends on the analysis to what extent we can still use the data. In some multivariate analyses —like multivariate normative comparisons for the cognition tasks— we can only use data from all non-missing instruments for a particular participant. For other analyses, like factor analyses, measurement invariance analyses, and linear mixed models, we can use Full Information Maximum Likelihood, to estimate models using all the available information. For the time-point level, we also use these models (linear mixed models) that can accommodate missing values. From our past experience, we do not expect data to be “Missing Not At Random”. How data will be imputed will be determined for each separate research question in the As Predicted registration as this will also be dependent on the analysis method used. We now briefly mention the difference ways needed to handle missing data on page 15 of the revised manuscript.

Page 15 “All final and detailed data analysis plans regarding future manuscripts based on the combined data across the different waves will be preregistered in detail on AsPredicted or similar platforms before data collection of Wave 4 ends. In these plans we will also include attrition rate handling, outlier detection, and how we will deal with item-level, instrument-level, or time-point-level missing data depending on the question at hand.”

d. Will you have power to look at different age groups? You match to control by age but if you wanted to see trajectories of 55+ are you powered to do that?

Re: To answer the reviewer’s question, yes, we do expect to have fewer very old participants (>70 years), so in that sense, we have lower power to detect differences between older participants with and without an autism diagnosis. However, we do expect larger ESs in older participants than in younger participants. For group comparisons, we perform, when applicable, our analysis both within a classical frequentist statistical framework and a Bayesian framework, so we can evaluate whether a possible lack of group differences is supported by the data, or whether it is better conceptualized as a lack of evidence/ sufficient data. To maintain power, we intend to discretize as little as possible, so we do not intend to always split our sample into smaller age groups. However, in order to compare our findings with previous findings we will explore whether differences emerge when comparing those below 55 and above 55 years of age. For our analysis of changes over time, we use for our main analyses the data of participants with T1 and T2 only.

2.2. So the aim 1 and 2 samples are different with aim 1 being a subset of aim 2? If that is correct, then you can’t use cognition variables in your trajectory analysis, since some won’t have those measurements? It would seem like they would be important variables to include, especially considering the epilepsy is so common.

Re: Our first aim is to arbitrate between cognitive aging hypotheses and our second aim is focused on subgrouping based on questionnaire data. To answer separate parts of this comment in turn, yes, the sample for aim 2 (subgrouping) is larger than for aim 1 (trajectories of cognitive aging), but the participants for aim 1 are a subset of those for aim 2. So indeed, we are not able to use cognition variables in the main subgrouping analysis, because some participants will not have cognitive data. The trajectory analysis is specifically focused on cognition, so for this analysis, we will use the cognition variables. Please see page 16 where we now explicitly state “as participants needed for Aim 1, will be a subset of the participants needed for Aim 2”

While epilepsy is indeed common in the general population of autistic adults, it will not be common in the subsample of participants for which we measured cognition. We do not include people with IQ levels below 80, and we exclude people for the cognitive session when major neurological problems are known. Hence, while based on the literature epilepsy seems to a potential candidate predictor, it

is not a likely predictor in the current study as we excluded those individuals with a self-reported history of epilepsy.

2.3. I found the figures more confusing than helpful. Can you add in calendar time to aid the reader in understanding the course of this project?

Re: We are sorry to hear this as we hoped that the figures would be illustrative given the complexity of the study design. We have now added calendar time (years in which each wave was performed) to the figures and excluded the information regarding ADHD. We hope that this aids in understanding.

2.4. It would be good to add how you will examine whether attrition is differential and how you will correct for that if it is the case. For example, if the people that don't show up for the visit don't show up because their health is so poor then that will bias your results.

Re: We thank the reviewer for this suggestion. Please note, we do our best to reduce systematic attrition by making participation easier. For example, we provide free parking, pick people up from the station, or have testing locations in people's home region (see also page 10). Despite this we will still examine the causes of attrition (see also page 17).

Page 15: "In these plans we will also include attrition rate handling, outlier detection, and how we will deal with item-level, instrument-level, or time-point-level missing data depending on the question at hand"

Page 17: "we will compare those individuals who were successfully followed up to those lost to follow up on the following variables: Autism and ADHD severity (based on AQ and ADHD-RS), age, estimated IQ, and the presence comorbid conditions (both psychiatric and somatic) in order to examine the impact of attrition on our findings. "

2.5. It is not entirely clear how this is an accelerated longitudinal design. Wouldn't you want to be recruiting more distinct age groups in each cohort? It seems like everyone after wave 1 would be 30-90, which is a very big range and will likely miss big gaps of the age range.

Re: In a typical accelerated longitudinal study each cohort starts at a different age when included. Here we do indeed deviate slightly from this classical accelerated design as we will be following a group of people from a broad age range over a variable time interval. However, our cohorts do start at a different age point, but overlap in the older age ranges. Where in Cohort 1 we recruited participants aged 18-80 (Wave 1), in Cohort 2 we recruited participants over 45 years of age (Wave 2), for Cohort 3 we are recruiting participants aged 30-90 years (Wave 3). Because of this, we will not have to follow each individual for a full lifetime, to be able to say something about development. Therefore, we believe that the current design should still be labeled accelerated longitudinal design.

On page 8 of the revised manuscript we added the following section: "Within in each of these three cohorts we start measuring at a different age point, but the cohorts do overlap in the older age ranges. Because of this, we will not have to follow each individual for a full lifetime, to be able to say something about development during adulthood. Essential is that within a relatively short time-frame [26], cognitive alterations can be detected from the age of 30 to 40 years onwards in healthy aging, where some of the cognitive processes start to decline from 30 onwards (i.e., processing speed), while others mainly decline after the age of 55 years (i.e., verbal memory) [27]."

In both Cohort 1 and Cohort 3 we recruited participants within age bins of 10 years of age to ensure that we do not have big gaps within this age range. In this way, we made sure that we had a similarly sized group of participants between 30 and 55 and those over 55 years of age and no apparent

gaps. So instead of following 30 year olds for a time span of 60 years, we follow 30-40, 40-50, 50-60 etcetera for 2 to 4 years. The only likely gap is of those individuals over 75 years as drop out might be higher for the oldest participants.

2.6. Similarly, what are you expecting in terms of your age distribution and gender distribution?

RE: We like to refer the reviewer to our previous comment regarding age. With respect to sex we recruited males:females in 2:1. We did not control the gender distribution (only sex), but did ask about people's gender identity. Please see also page 11 of the revised manuscript regarding the recruitment of participants across age bins and sex. "In order to ensure an equal proportion of participants across the whole age range and in order to match the comparison group with the ASC group, people are recruited within age bins of 10 years of age and with a sex distribution of 2:1 (males:females)."

2.7. Outside of analysis, the paper needs a discussion of why there have been so few studies of older autistic adults and how that impacts this study. Simply, we are not good at identifying autistic adults as having autism. There is no gold standard measure. Further, there are major cohort effects in autism, as someone that was diagnosed in 2016 at 6 years old will have a world that is very different towards autism as compared to someone diagnosed in 2016 at 60. I think that there needs to be some acknowledgement of all the external factors that impact a person on the spectrum and how those may drive cognitive trajectories or subgroups and how that may be confounded by cohort (as in age-period-cohort analyses). What does it mean that you are excluding those without a past ASD diagnosis? Does that exclusion make them more or less similar to controls?

Re: There are indeed several reasons why there are fewer studies in old age autistic adults and indeed there might be cohort effects. As for the protocol paper we have only a limited number of words, we decided to focus less on all the aspects that might be a reason for why people are only recently starting to focus on old age and aging as there are extensive reviews on each of these issues. We refer to these reviews on page 4 of the manuscript as follows "for an overview of reasons for the lack of aging research see [10,11]" Moreover, there are actually large differences across countries in adult autism diagnoses. Adult diagnoses are common in the Netherlands, the UK, and most Scandinavian countries for at least 20 years. For example, two of the authors work in an autism clinic which has a specialist team which started 20 years ago and a so called workhome since the late seventies for autistic adults. Nonetheless cohort effects will be accounted for in the analyses by adding cohort as factor to the linear mixed models.

With respect to the exclusion, there might be a misunderstanding. We do not exclude people without a past ASC diagnosis in the ASC group, we are excluding the people who do not have an official clinical ASC diagnosis in the ASC group. While it is of interest to see how undiagnosed (but self-identifying as autistic) individuals age, this is outside the scope of the current study. Unfortunately, we had to kill some darlings in order to make the study feasible. It seems more likely that most participants in our study will have had their diagnosis in the last 5 to 20 years. As we record when they were first diagnosed with ASC, we will have this information so we can explore whether age-of-diagnosis and era-of-diagnosis has an impact on our outcomes.

10 Happé F, Charlton RA. Aging in autism spectrum disorders: a mini-review. *Gerontology* 2012;58:70–8. doi:10.1159/000329720

11 Lai M-C, Baron-Cohen S. Identifying the lost generation of adults with autism spectrum conditions. *Lancet Psychiatry* 2015;2:1013–27. doi:10.1016/S2215-0366(15)00277-1

2.8. Subgroups in autism tend not to replicate. Consider tempering the paragraph in the conclusion about using the subgroups as they will need to be replicated using external data before clinical changes should take place.

Re: The editor has requested us to remove the conclusion section, see above. However, please note that in our study we test whether we can replicate our subgrouping. So, we use a T1 training set (Wave 2), a T1 test set (Wave 3), and the subgroups can be tested again at T2 (Wave 4). Whether these subgroups can also be replicated by others is for the future. As we recently reviewed all subtyping papers since 2000 in ASC (paper submitted), we are indeed well aware that there seem to be no valid subgroups yet. However, this does not mean that we will not be able to establish valid subgroups in our study, as in earlier studies not all methodological options have been used to their full potential. In our now more extensive statistical analysis section we highlight the different validation methods we use including replication (please see also comment #2.1, and page 8 and 15 of the manuscript).

Page 8 "we can test specificity, external validity, temporal stability, predictive validity, and replicability for behaviorally defined subtypes in autistic adults."

Page 15 "We also provided a detailed analysis plan for subgrouping on T1 Wave 2 and Wave 3 data by means of community detection at <https://aspredicted.org/blind.php?x=hu4ey6>."

2.9. The word senescence does not appear anywhere but the title. I had to look up what it meant after I read the paper because I was confused not seeing it in the text.

RE: Please see our reply (#1.1) to reviewer 1. In short, we adjusted the title and replaced senescence by aging.

Reviewer: 3

Reviewer Name: Stephen M. Edelson

Comments to the Author

The manuscript titled "Senescence and heterogeneity regarding autism spectrum conditions: A protocol paper of an accelerated longitudinal study" [bmjopen-2020-040943] describes a much-needed research project that will likely provide much insight about adults on the autism spectrum as they age into their senior years.

Re: We thank the reviewer for this compliment, our replies to your comments are below.

I have listed my comments below:

3.1. The authors discuss two aging hypotheses, i.e., accelerated cognitive decline and protective cognitive strategy use. Another possible hypothesis often reported in those who are developmentally delayed or mentally challenged is a gradual maturation (i.e., improvement) in cognitive function over time.

Re: Perhaps not made sufficiently explicit, we have a third alternative (the null hypothesis): aging proceeds in a parallel fashion, so similar to typical aging, but with a different starting point (now mentioned on page 5 of the revised manuscript). However, it is indeed true that maturation could be shifted in time and thus also aging. So the development and deterioration in cognitive functioning might have a similar distribution in ASC and COM, but is just shifted in time (i.e., the cognitive performance distribution is shifted to the right along the x-axis, so the peak of maturation is at a later age for the ASC group as compared to

the COM group). We now mention this briefly on page 5 (“While the process of cognitive decline might be similar for autistic adults and non-autistic adults (i.e., parallel aging or delayed aging where cognitive maturation and cognitive decline are shifted in time but follow a similar pattern)”) and page 6 (“and delayed aging”) of the revised manuscript, but below we describe why we believe that it will be difficult to properly test this alternative hypothesis due to the large number of unknowns.

Delayed aging would imply that cognitive decline sets in later in time in autistic adults when compared to people without ASC. Factors such as where across the life span the peak of maturation is reached, whether or not this is before or after 30 years of age, and whether the peak is similar in height and/or width for both groups will all impact the type of pattern one might expect.

If the peak of maturation is before 30 in both groups one would observe a parallel aging pattern (both linear) in the time span we are measuring the ASC group would outperform the COM group, at each age. In cross-sectional studies so far, we and others did not observe that the ASC group is outperforming the COM group across the age-range we are focusing on. So if both groups peak before age of 30, the alternative hypothesis of delayed aging seems unlikely based on previous findings.

However, the proposed hypotheses could be an alternative explanation if the peak for the COM group is before 30 (which is in line with the typical development literature for all cognitive processes we focus on) and for the ASC group close to 30 years of age. In this scenario, one would expect a predominantly linear relation in the COM group and a quadratic relation in the ASC group during the first couple of years we measure (>30 years). Here we would observe an interaction where the COM group first might outperform the ASC group while at an older age (e.g., 60+) both groups show a similar performance or the ASC group outperforms the COM group. This implies that the ASC group would outperform the COM group at each age after the ASC performance peak has been reached. A pattern which we labelled as protective, but could in theory indeed also be caused by delayed maturation. In this case the shape of the relationship between performance and age, combined with what we know about early development, would need to be leading in determining which explanation is the most plausible. The question is whether, based on what we currently know about cognitive development in autistic people, it is plausible to assume that their maturation peak is reached after the age of 30 for the cognitive processes we study.

In sum, it is a valid alternative hypothesis, but it will be hard to disentangle from any of the others and to match this hypothesis with the observations in the literature so far. However, we will definitely keep this in mind for our current and future thinking, and have, as aforementioned, incorporated it in our manuscript on page 5.

3.2 It is not clear whether early developmental events will also be collected. Some critical events will likely be well-remembered, such as pregnancy and her birth complications, age began walking and talking, and the presence of seizures.

Re: Although we agree with the reviewer that these can potentially be important, early developmental events will not be collected in this study. Retrospective reporting becomes more unreliable with increasing age, and we, unfortunately, could not measure all aspects which might be of interest. We also had to take feasibility into account and decided to first focus on more recent events.

3.3 Will there be any attempt to document, in relation to age, whether the participants received intensive behavior therapy, sensory/OT interventions, medications, and so on? It is quite possible that certain therapies, especially administered during childhood, could determine a range of developmental trajectories in relation to each subtype.

Re: It is indeed a very interesting question to assess whether those who were treated at an early age have a different developmental trajectory compared to those who were not treated at an early age. While we do collect data on medication use, we will not collect data on other types of treatment, and we will thus, unfortunately not be able to answer this relevant question.

3.4 I suggest that the authors consider administering a questionnaire designed specifically for sleep disturbances associated with autism. Sleep is quite prevalent among those on the spectrum, and quality of sleep may impact cognition (e.g., inattention, memory retrieval) and behavior (irritability, aggression).

Re: The current study is already expected to be quite taxing on the participants, and we could therefore not include measures that were not part of our main aims. However, there are 3 questions in the SCL-90, and 1 question in the WHO-QOL-BREF on sleep.

3.5 Page 12 lines 26 and 36. The authors refer to the COM group. What does this refer to? Earlier they use the term COMP to refer to a comparison group (page 10, line 24).

Re: We thank the reviewer for detecting this typo, we have corrected it and now use COM throughout the manuscript. We also added the following information on page 8 “an ASC group, a typically developing comparison group (COM), and an ADHD group”.

3.6 Will there be any discussion with the participant and/or family members regarding postmortem brain research?

Re: We do not discuss postmortem brain research with the participants. However, when participants have questions about this topic, we will guide them to the Dutch brain foundation.

I commend the authors' efforts in undertaking such an exciting and important research project.

Re: Thanks, we are really looking forward to analyze the data in order to see what is going on in old age.

VERSION 2 – REVIEW

REVIEWER	Eric Rubenstein Boston University, USA
REVIEW RETURNED	05-Jan-2021

GENERAL COMMENTS	Thank you for addressing all my concerns. The protocol paper looks great and I can't wait to read about your findings.
--

REVIEWER	Stephen M. Edelson Autism Research Institute, USA
REVIEW RETURNED	20-Jan-2021

GENERAL COMMENTS	The authors did a fine job revising the manuscript. These findings will likely provide important and needed insight on aging in autism.
---